# Advancements and Prospects of Genome-Wide Association Studies (GWAS) in Maize

**DOI:** 10.3390/ijms25031918

**Published:** 2024-02-05

**Authors:** Javed Hussain Sahito, Hao Zhang, Zeeshan Ghulam Nabi Gishkori, Chenhui Ma, Zhihao Wang, Dong Ding, Xuehai Zhang, Jihua Tang

**Affiliations:** 1National Key Laboratory of Wheat and Maize Crop Science, College of Agronomy, Henan Agricultural University, Zhengzhou 450002, China; 2Institute of Biotechnology, College of Agriculture and Biotechnology, Zhejiang University, Hangzhou 310058, China; 3The Shennong Laboratory, Zhengzhou 450002, China

**Keywords:** GWAS, maize, candidate genes, agronomic traits, environmental stress

## Abstract

Genome-wide association studies (GWAS) have emerged as a powerful tool for unraveling intricate genotype–phenotype association across various species. Maize (*Zea mays* L.), renowned for its extensive genetic diversity and rapid linkage disequilibrium (LD), stands as an exemplary candidate for GWAS. In maize, GWAS has made significant advancements by pinpointing numerous genetic loci and potential genes associated with complex traits, including responses to both abiotic and biotic stress. These discoveries hold the promise of enhancing adaptability and yield through effective breeding strategies. Nevertheless, the impact of environmental stress on crop growth and yield is evident in various agronomic traits. Therefore, understanding the complex genetic basis of these traits becomes paramount. This review delves into current and future prospectives aimed at yield, quality, and environmental stress resilience in maize and also addresses the challenges encountered during genomic selection and molecular breeding, all facilitated by the utilization of GWAS. Furthermore, the integration of omics, including genomics, transcriptomics, proteomics, metabolomics, epigenomics, and phenomics has enriched our understanding of intricate traits in maize, thereby enhancing environmental stress tolerance and boosting maize production. Collectively, these insights not only advance our understanding of the genetic mechanism regulating complex traits but also propel the utilization of marker-assisted selection in maize molecular breeding programs, where GWAS plays a pivotal role. Therefore, GWAS provides robust support for delving into the genetic mechanism underlying complex traits in maize and enhancing breeding strategies.

## 1. Introduction

Maize (*Zea mays* L.) stands as a cornerstone in global agriculture, playing an irreplaceable role as a fundamental staple for food, feed, and fodder. Beyond its vital contributions to sustenance, it also holds economic significance across industries encompassing beverages, paper, pharmaceuticals, and textiles [1,2,3]. However, maize confronts a diverse spectrum of both biotic and abiotic stresses throughout its developmental stages. These challenges encompass a range of factors including drought, salinity, thermal extremes, cold stress, waterlogging, diseases, and nutrient in-sufficiency [4,5,6]. Biotic and abiotic stresses exert an adverse impact on approximately half of the global crop yields [7]. Therefore, enhancing productivity and biotic and abiotic resistance in maize varieties becomes a central objective, encompassing traits such as seed germination, root and shoot development, photosynthesis, osmotic characteristics, and cereal plant architecture through the development of new and improved technologies followed by their adaptation and popularization. These traits are crucial for primary growth and survival under stress conditions [8,9,10,11].

Simultaneously, unraveling the intricate mechanisms of biotic and abiotic resistance is essential to ensuring maize production on a global scale. Conventional breeding methods have introduced numerous maize varieties. However, these approaches often fall short of meeting the demand for increased yield and enhanced biotic and abiotic stresses tolerance. In recent years, there has been a shift towards exploring diverse methodologies aimed at understanding the intricate relationship between genotypes and multiple important traits. Among these, quantitative trait loci (QTL) mining techniques, such as QTL mapping and association mapping, have gained prominence [3,12]. Furthermore, association mapping encompasses two primary categories: genome-wide association studies (GWAS) and candidate gene association mapping [13]. Scholars have elucidated the distinguishing features between QTL mapping and GWAS in their studies [14,15].

However, QTL mapping encounters limitations in effectively deciphering traits controlled by numerous minor-effect QTL in maize. For instance, traits such as resistance to the fungal disease Fusarium ear rot display complexity, with linkage mapping identifying only a few QTL in each study [3,15]. Additionally, the extensive and intricate maize genome, consisting of 85% repetitive sequences, poses challenges to precise QTL mapping and cloning [15,16]. QTL mapping is characterized by restricted allele frequency, reduced speed, and limited resolution power [12]. In contrast, GWAS offer a more robust approach, providing enhanced resolution, higher allelic frequencies, comprehensive genome coverage, and the ability to detect numerous historical recombination events [3,14,15,17,18] (Figure 1). GWAS have revealed genomic regions associated with numerous physiological, agronomical, and fitness traits. These traits span a range of characteristics, such as plant height, flowering time, kernel number, stress tolerance, and grain yield in plants [19]. GWAS is not only performed in maize but also successfully applied to investigated complex traits in many plant species including Arabidopsis, rice, soybean, wheat, cotton, sorghum, etc. [20,21,22]. In addition, a number of reviews have been published on GWAS in plants [3,21,23,24,25,26,27]. Moreover, GWAS have identified genomic regions associated with a number of agronomic traits, physiological, biochemical and cellular traits, and fitness traits such as plant height, flowering time, number of kernels, abiotic and biotic stresses tolerance, grain yield, etc. [19]. More than 20 years, an initial GWAS was performed in Arabidopsis focused on SNPs, recombination, and LD [28,29]. In Arabidopsis, the functional gene ACCELERATED CELL DEATH6 (ACD6) was identified thorough GWAS, indicating a trade-off between metabolism and defense [30,31]. In 2010, the first GWAS was performed in rice for 14 agronomic traits, in which a high-density haplotype map was created using a skim-sequence of 517 rice lines with genotypes imputed by the algorithm k-nearest neighbor (KNN) method [32,33]. Taken together, these studies provide unprecedented resources for a better understanding of functional genomics in plants.

Maize’s extensive genetic diversity, the abundance of single nucleotide polymorphisms (SNPs), the decay of linkage disequilibrium (LD), and the existence of diverse sub-populations collectively establish it as an ideal candidate for GWAS [3,15,34,35]. The first GWAS in maize investigated genes influencing the fatty-acid content in maize kernels, utilizing 8590 loci in 553 elite maize inbred lines [36]. Over the past decade, significant progress has been achieved in maize GWAS. GWAS have emerged as a powerful tool for elucidating associations between SNPs, QTL, and candidate genes associated with complex traits in various plant species. This approach has been further strengthened by the advancements in next-generation sequencing (NGS) technologies and releasing of the B73 reference genome [37].

This review encompasses three critical dimensions: (i) recent advancements in maize GWAS and their fundamental methodology, (ii) future prospects, and (iii) challenges in functional genomics, along with the compelling questions that will shape the trajectory of maize molecular breeding programs.

## 2. GWAS in Maize

GWAS, also known as genome-wide association studies, have emerged as a powerful tool that leverages phenotypic and genotypic variations within plant species to identify favorable alleles associated with desired traits. In the context of maize research, GWAS haves been extensively applied to investigate the relationship between genetic markers and phenotypic traits. The first GWAS in maize was conducted in 2008, targeting the identification of SNPs that significantly affect the oleic acid content in kernels by utilizing 8590 loci in 553 elite inbred lines [36]. Since then, accompanied by the release of the “B73” reference maize genome [15,16], GWAS have undergone notable advancements and have become a common technique for uncovering genotype–phenotype relationships in maize. It offers a powerful method of identifying the functional genomics of genes. Previous studies have successfully identified candidate genes responsive to both biotic and abiotic stresses in maize, providing valuable insights [3,15,18]. In addition, GWAS have enabled the discovery of numerous putative candidate genes associated with various traits in maize, further enhancing our understanding of its genetic architecture. GWAS allow for the fine mapping of QTL, utilizing a diverse maize population, confronting a huge number of historical recombination events which may result in the rapid LD decay, and the mapping precision can reach the single-gene level [38].

GWAS in maize are characterized by the rapid LD decay due to the crop’s diverse genotypic and phenotypic characteristics compared to other species [12]. Some genes do not segregate independently because the two loci involved are located on the same chromosome. Simply stated, LD, a measure of the non-random association of alleles at different loci, can be estimated using metrics like Lewontin’s (‘D’) and coefficient of determination (‘C’) analyses [3,39]. LD strength between two SNPs is often quantified using the ‘r^2^’ estimate, if r^2^ values are below 0.2 indicating co-inheritance and two SNPs are present on the same QTL [14]. Lower levels of LD enable higher-resolution association mapping, emphasizing loci significantly associated with the interested trait [15]. However, achieving such resolution requires a larger number of molecular markers compared to situations with higher LD. The higher probability of LD in plants also facilitates the identification of markers associated with causal variants [38]. GWAS have been successful in identifying genomic regions and QTL that control disease tolerance in maize, such as southern maize rust, leaf necrosis, gibberella ear rot, fusarium ear rot, and gray leaf spot [3]. GWAS offer a rapid method of unraveling the genetic basis of complex traits, outperforming traditional linkage mapping in terms of resolution and cost-effectiveness [3,40]. Moreover, GWAS have been highly successful in identifying numerous genomic regions associated with various abiotic stressors, indicating genetic diversity for morphological and physiological characteristics across diverse maize populations [41,42]. Maize, with its abundant marker density and high-density genotyping technologies, is an ideal cereal crop for GWAS. This method has recently emerged as a critical approach for studying natural variation and mapping quantitative traits, enabling the detailed exploration of genetic architecture in maize. Utilizing high-resolution genotyping technologies such as the IlluminaTM 9 k SNP chip, GWAS have the potential to uncover novel alleles that enhance production and adaptability in maize [14,43].

## 3. The Basic GWAS Approach

GWAS serve the purpose of identifying associations between genotypes and target traits. It has emerged as a powerful alternative to conventional QTL mapping for pinpointing the genetic loci underlying traits with a high resolution [44,45]. The Basic GWAS Approach is shown in Figure 2; when genotypes and phenotypes are obtained, GWAS analysis can be performed, in addition, to improve the mapping power, and several statistical methods in GWAS have been employed with the corresponding software Tassel version 5.0.

The standard MLM is less effective for large datasets with hundreds of individuals due to computational overhead required for numerical optimization [3]. These models are typically designed for single-locus testing, but effective multi-locus mixed models have also been developed for GWAS [46]. While SNP-based GWAS are widely used, they can suffer from weak signals within related SNP sets and overlook SNP interactions within a gene [47]. To address these issues, haplotype-based GWAS and gene-based GWAS have been introduced, offering high statistical power to identify causal haplotypes and uncover new candidates for complex traits, especially for rare alleles [48]. These statistical methods generally assume that phenotypes and noticeable effects follow a normal distribution. For complex traits, fine mapping association methods based on statistics have been developed [49]. The Anderson–Darling test is particularly useful for identifying loci with modest effects and rare variants that exhibit aberrant distribution of phenotypes. It offers a balanced approach in terms of false positives and statistical power when compared to mixed models [50]. GWAS have proven successful in identifying common disease variants and revealing new sensitivity loci for various complex diseases, shedding light on their unknown functions. While initial concerns arose regarding the small effect sizes of GWAS loci and their partial explanation of missing heritability [51], these limitations are minimal, especially when large and diverse populations are used. There seems to be a sample-size threshold at which the discovery rate in GWAS accelerates for each complex trait. Larger populations and more SNPs increase the likelihood of successful discoveries [52,53].

On the other hand, the most challenging situations arise when identifying genes associated with phenotypic variation in traits with high genetic complexity and low heritability. This occurs when phenotypic variation is influenced by environmental factors, and the genetic basis of the complex trait involves numerous genes, each making a small contribution to the genetic variance of the trait [53]. It is important to note that the value of biological understanding gained through GWAS is not solely tied to the strength of association, which underscores the need for identifying minor associations in larger sample sizes [48]. Various methods exist for genotyping genetic variants, with whole-genome sequencing (WGS) and SNP arrays being the most common. While SNP arrays are less expensive and offer precision and a well-established analysis process, WGS, despite being costlier and less accurate, can detect rare variations effectively, especially with large sample sizes. Imputation from sequencing data to SNP array data can be highly reliable with a large sample size [54].

In summary, both WGS and SNP arrays continue to be effective methods, and the choice between them depends on factors such as cost, precision, and research goals. Currently, WGS offers greater potential for resolving limitations and explaining missing heritability, especially with advances in genotyping technologies [48,55]. In addition, missing heritability in GWAS defines a significant issue in the genetic analysis of complex traits and identifies the causal genetic variants and quantifies their individual contribution [56,57]. The missing heritability can be estimated through genetic studies because of the use of fast-developing genomics for better understanding biological problems and crop breeding [57,58]. Recently, graph pangenome was used for GWAS to capture missing heritability in tomato breeding in order to understanding the heritability of complex traits and showed the power of graph pangenome in crop breeding [57]. GWAS are frequently using WGS and whole exome sequencing (WES) to enable the identification of rare variants, which could explain the missing heritability in complex traits [59]. These strategies can be considered to address the challenges of missing heritability through the utilization of GWAS, but the strategy depends on the interested trait and its adaptive scale. Additionally, two independent methods were developed: PEPIS for multiple hybrid populations (MHPs) corresponded with compressed mixed linear model (CMLM) to discover candidate genes and QTNs for maize in GWAS [60]. Moreover, GWAS was conducted using a maize heterosis population for genetic variation among populations, and three models were implemented, namely, GLM, MLM, and FarmCPU [61], and an additive genetic model was used for per se trials, whereas dominant models were used for test cross trials. Nevertheless, additional algorithms in the R program GAPIT were used to performed GWAS for each model [62,63,64]. Indeed, mixed models have become a standard approach for analyzing large datasets, utilizing both phenotypic and genotypic data to improve the resolution power of GWAS. However, it is important to note that these approaches may render the earlier methods computationally unfeasible due to increased complexity. Consequently, there is a need for the development of new statistical methods that are both statistically robust and computationally efficient to overcome these challenges. The general approach and various methods can be applied to a wide range of GWAS analyses.

## 4. Factors Affecting the Accuracy and Statistical Power of GWAS

GWAS possess a remarkable ability to uncover precise associations between genetic markers and phenotypic traits. However, the effectiveness of GWAS is influenced by various factors, particularly when it is applied to specific target traits. A key determinant in this regard is the extent of phenotypic trait variance within the natural population. The power of GWAS is closely linked to its resolution, as a higher resolution enhances the accuracy of identifying candidate genes associated with a particular trait. Notably, it also aids in the discovery of novel genes associated with the trait of interest [65]. In the realm of maize genetics, GWAS has played a pivotal role in identifying thousands of candidate genes linked to important traits. However, the challenge lies in understanding how much phenotypic variance can be attributed to the variations in two alleles, and this often hinges on the magnitude of their influence and their frequency differences within the sample. It is worth noting that GWAS face impediments in the form of rare variants and small effect sizes when attempting to elucidate these associations [37]. In Figure 3, we delve into four crucial factors that significantly impact the accuracy and power of GWAS, as detailed below [14]:

**(1) Phenotypic variation in a population:** Phenotypic variation plays a pivotal role in the association analysis, with the removal of outliers having a relatively minor impact. To facilitate further analysis, it is essential to filter raw phenotypic data to eliminate outliers and noisy data points. The use of a boxplot can help identify and assess the relevance of outliers within the phenotypic data. It is worth noting that retaining extreme outliers may deviate the phenotypic data from a normal distribution, potentially limiting the scope of GWAS. Additionally, for effective GWAS, it is advisable to employ traits with moderate-to-high heritability estimates (for phenotypic data filtration). High heritability serves as a reliable indicator of the extent to how much genetic variation influences phenotypes, thus establishing a strong connection between genotype and phenotype. In contrast, low broad-sense heritability is a limiting factor that diminishes the power of GWAS and the ability to detect associations. Genotype–environment interactions may reduce heritability of a trait when genotypes are tested across diverse locations or years, but various methods, such as the best linear unbiased predictor (BLUP) and best linear unbiased estimator (BLUE), can be employed to modify phenotypic data scored across different locations or years, thereby providing more accurate estimates of phenotypic values while considering genotype–environment interactions [15].

**(2) Population size:** The population size is a fundamental determinant in GWAS, affecting both phenotypic and genotypic variance. Increasing the population size enhances the power to identify significant associations, especially those with substantial effects and reasonable frequencies within the population, thereby overcoming rare variants, which are those with low allele frequencies. Conversely, a smaller population size of individuals is a disadvantage that diminished the power of GWAS [66,67]. Careful selection of individuals based on expected phenotypic and genotypic variation, and considering genetic background factors such as environmental regions, biological status, growth habits, or other relevant traits can help mitigate this limitation. In general, a larger population size, ideally ranging from 100 to 500 individuals, is recommended for optimal GWAS outcomes [15].

**(3) Population structure:** Population structure is a statistical approach used to calculate the relatedness correlation among individuals within a population, taking into account factors such as admixture and historical structure. It is crucial to carefully consider population structure during the analysis and interpreting of GWAS results. Researchers typically select populations for GWAS based on factors like growth habit, geography, etc., which can generate the population structure. This, in turn, leads to a specific genetic variant and can influence the final results of the association analysis. The main fundamental limitation of GWAS lies in the fact that not all individuals within a population are genetically equally distinct from each other. Ignoring this to account for population structure can result in a spurious association between the genotype and interested traits. Therefore, it is essential to incorporate robust statistical methods to control for the population structure in order to obtain accurate and meaningful results from GWAS. Employing a mixed-effect model that accounts for population structure can alleviate this issue [66]. Software like STRUCTURE V2.3.4 assists in defining the population structure, calculating the proportion clusters (Q matrix), and estimating individual sub-population membership using genotype multi-locus data. Most of the past research employed both methods, STRUCTURE and Principal Component Analysis (PCA), to derive their results [14,35,68]. PCA offers another method for estimating the population structure efficiently, often outperforming STRUCTURE methods at the time of calculation [13].

**(4) Linkage disequilibrium (LD):** LD refers to the non-random association of alleles at different loci, a crucial factor for association mapping studies [39,69,70]. High LD values indicate that fewer markers are required to cover the genome, while calculating LD at the outset of association studies is crucial to avoid false associations [71]. Various statistics, including r^2^ and D’, are used to measure LD [13]. Moreover, the LD decay rate across the distances is crucial for determining the number of markers in GWAS. Self-pollinated crops typically display a larger LD decay compared to cross-pollinated crops, necessitating fewer markers to cover the entire genome [72]. Structural analysis of LD can serve as an initial step in designing GWAS investigations.

## 5. GWAS on Agronomic, Quality, and Quantitative Traits in Maize

GWAS have become a powerful tool for unraveling the intricate genetic architecture of complex traits in maize, shedding light on the presence of QTL and identifying candidate genes responsible for these traits [73]. Specifically, GWAS have played a pivotal role in elucidating the connections between SNPs and quality traits, thereby deepening our comprehension of the genetic basis of these characteristics [74]. The emergence of next-generation sequencing (NGS) has catapulted GWAS to the forefront of maize genetics research, marking a significant milestone in the field [75]. Over the past decade, substantial strides have been taken in this domain. For instance, a seminal study identified a total of 74 loci associated with oil biosynthesis and fatty acids in maize kernels. This research attributed more than 10% of the variability in kernel oil concentration to various genes [76]. Furthermore, maize research has identified 29 candidate genes and 49 SNPs linked to grain quality, with these grain-related traits demonstrating a notably high broad-sense heritability [75]. In another groundbreaking study, researchers successfully mapped more than 40 QTL, unveiling a multitude of QTL with small effects. These QTL were discovered under a straightforward additive model, enabling the prediction of flowering time in maize. This remarkable achievement was accomplished using a vast population of five thousand individuals nested within an association mapping framework [77]. In yet another significant investigation, researchers delved into associated loci that elucidated over 20% of the observed variance in secondary metabolism traits. These traits exhibited a median effect size of 7.8% in maize kernels, underscoring the substantial impact of these loci on maize genetics [78].

Similarly, 59 SNPs were identified to elucidate the genetic basis of maize yield-related traits, including grain yield per plant, grain width, grain length, 100-kernel weight, number of kernels per row, and tassel branch number [79]. Zheng and colleagues identified 49 SNPs and 29 candidate genes related to kernel quality in maize through GWAS [75]. Likewise, GWAS were conducted to explore the genetic basis of kernel-related traits, detecting a total of 139 SNPs and 15 genes enriched in regulating oxidoreductase activity, leaf senescence, and peroxidase activity [80]. Moreover, 46 SNPs that exhibited significant associations with zinc and iron contents in maize kernels were unveiled [39], shedding light on the genetic factors influencing these essential micronutrients. For forage quality traits such as acid detergent fiber, neutral detergent fiber, and in vitro dry matter digestibility in diverse maize, 73, 41, and 82 SNPs were found to be associated, respectively [81]. Additionally, 18, 22, and 24 SNPs significantly linked to cellulose, hemicellulose, and lignin in maize were uncovered [15], enriching our understanding of the genetic basis of cell wall composition in this crop. These studies also unveiled candidate genes involved in various biological pathways, including cell wall metabolism and protein kinases [82]. In a recent study [83], the authors identified a total of 48 SNPs and 37 candidate genes associated with starch pasting properties and maize kernel quality traits, further expanding our knowledge of maize kernel traits. The continued growth of GWAS indicates its increasing significance in maize genetic research, emerging as a pivotal tool for unraveling the genetic mechanisms governing agronomic traits and key phenotypes [3,84,85]. To date, GWAS has explored a wide array of traits, spanning molecular, cellular, agronomic, quality, and quantitative traits, while also considering interactions with biotic and abiotic factors. These research endeavors have culminated in the identification of numerous candidate genes associated with various maize traits. Future studies should prioritize the functional validation of candidate genes and field evaluations of germplasm harboring causal or associated SNPs and genes. These efforts hold promise for generating abiotic stress-resistant maize genotypes with high-yield characteristics, contributing to the advancement of maize agriculture.

## 6. Factors Affecting Maize Production

Maize stands as one of the foremost cereal crops, with grain yield emerging as a critical composite trait vulnerable to an array of environmental factors. These factors, comprising both biotic and abiotic stresses, pose a significant threat to maize cultivation [42,48,86]. Plants have developed intricate molecular pathways to contend with these environmental challenges. In recent years, the constantly shifting climate conditions have led to an escalation in the frequency and intensity of extreme weather events, resulting in substantial crop losses. In parallel, the global expansion of trade has facilitated the spread of disease-causing pathogens to new regions, where they adapt, giving rise to persistent outbreaks [87]. Simultaneously, insect pests and pathogenic diseases can assail maize at various stages of growth, affecting both grain yield and quality. Researchers have conducted numerous studies, extensively exploring these biotic stresses in the field [88,89,90,91]. Abiotic stresses, on the other hand, encompass adverse environmental conditions such as drought, heat, chilling, flooding, and salinity [92,93,94,95,96,97]. These stresses not only hinder plant growth and development but also impact nutritional composition, ultimately influencing maize grain quality. Recognizing and evaluating a plant’s ability to withstand both biotic and abiotic stress is a pivotal step towards enhancing maize productivity [81]. In maize, the genotypic diversity regarding stress tolerance can be assessed through appropriate screening processes, alongside the identification of traits correlated with stress resilience [98]. Hence, screening for stress resistance has relied on potential traits, including plant height, anthesis–silking interval, ear height, and grain yield components under stress conditions. Furthermore, certain characteristics exhibit strong heredity and high relevance to stress tolerance, such as brace roots for water stress, stay-green traits, tassel sterility, silk balling, and tassel blast, which are affected by heat stress [15]. These traits hold promise for identifying stress-tolerant maize varieties in future molecular breeding programs.

Understanding the molecular mechanisms governing plant stress responses is pivotal in developing climate-resilient maize varieties [99]. GWAS have proven invaluable in elucidating the intricate genetic architecture, susceptibility loci, and candidate genes involved in both biotic and abiotic stress pathways. In the following sections, we will delve into select studies that employ GWAS to unravel the genetic underpinnings of biotic and abiotic stress resilience in maize, as conducted by other researchers. Biotic stress, typically induced by diseases or insect pests, ranks among the primary contributors to maize yield losses. Several diseases have been identified in maize, including turcicum leaf blight, maize rough dwarf disease, ear rot, aflatoxin contamination, and sugarcane mosaic disease.

In addition to the biotic stresses mentioned earlier, maize cultivation faces threats from various pests, including stem borers, rootworm, pink borer, shoot fly, storage pests like weevils, and termites [100,101]. Addressing these challenges is essential to safeguard maize production. Initially, researchers utilized nested association mapping populations to identify 32 QTL with subtle additive effects on southern leaf blight resistance in maize. Notably, many of these QTL were found to be located near or within genes previously implicated in plant disease resistance [102]. The use of GWAS has enabled the creation of molecular markers, facilitating the indirect selection of traits such as resistance to corn earworm, which is otherwise challenging to achieve through conventional breeding programs [103]. Moreover, GWAS employing gene-set enrichment approaches have been instrumental in identifying groups of genes that collectively contribute to resistance. For example, 4 loci and 16 candidate genes were discovered through GWAS in maize kernels, conferring resistance to Aspergillus flavus fungal disease in grains [104]. Another GWAS identified 14 SNPs associated with resistance to northern corn leaf blight caused by Exserohilum turcicum [105]. GWAS have been instrumental in pinpointing genomic loci and allelic variants that govern resistance to maize lethal necrosis [106]. Certain maize lines, such as dent and flint, inherently resist gibberella ear rot, a fungal disease caused by Gibberella pathogens [107]. A GWAS conducted by [108] identified the major QTL qRtsc8-1, accounting for a substantial portion of the observed variation in tar spot resistance (18% to 43%). In another study [109], the authors utilized GWAS to identify 22 SNPs and two candidate genes linked to resistance against maize rough dwarf disease.

These findings have shed light on candidate genes encoding proteins and enzymes involved in signal transduction, stress response, and various aspects of transcriptional and post-transcriptional control of cell component synthesis. They hold significant promise for molecular breeding in maize to enhance disease resistance and advance our understanding of the genetic basis of resistance in maize. Similarly, abiotic stresses exert a considerable impact on maize growth and yield, especially in the context of ongoing climate change. Maize’s tolerance to abiotic stressors like drought, heat, salt, cold, and water submersion has been investigated extensively through GWAS. This approach has offered insights into potential SNPs and candidate genes for enhancing maize yield. For instance, the gene *ZmVPP1,* encoding vascular pyrophosphatase, has been found to improve drought tolerance in maize seedlings [110], and GWAS have identified ten loci associated with metabolites linked to drought tolerance [85].

In the context of cold stress, genes like *ZmACA1*, *ZmDREB2A*, *ZmERF3*, and *ZmCOI6.1* have been identified in maize [111]. GWAS have also uncovered 24 *ZmFKBP* genes involved in multiple signaling pathways during stress [112], as well as *ZmPP2C2* and *ZmMKK4* [113,114]. Low-phosphorus-responsive genes have been identified in maize seedlings through GWAS, along with 259 genes associated with phosphorus stress tolerance [115]. Additionally, four QTL and candidate genes have been linked to thermotolerance in maize seeds [116], and *Zm00001eb198930* has been identified as responsive to high salt tolerance [117]. Chilling stress has also been a focus of GWAS, with the authors of [118] identifying 19 genes highly associated with early growth and chlorophyll fluorescence traits under chilling stress in field conditions. A recent study [119] highlighted three genes associated with chilling tolerance during maize germination. While GWAS have proven valuable in uncovering genetic factors related to stress tolerance, it is important to note that environmental factors play a significant role in gene expression. Thus, the results of GWAS should be validated under field conditions to account for the influence of shifting environmental factors. Field phenotyping can be challenging due to stress heterogeneity, varying plant responses, and simultaneous stress effects. Furthermore, GWAS have successfully identified several candidate genes associated with common abiotic and biotic stressors in maize, thereby revealing the genetic basis. For instance, a GWAS conducted under field conditions identified eight SNPs and favorable alleles associated with kernel moisture content in maize [120]. Additionally, GWAS were carried out to elucidate the genetic basis of various phenotypic traits at the sequence level in maize under field conditions [81,121]. Thousands of genomic regions associated with agronomical traits have been identified through the application of GWAS under both biotic and abiotic stress conditions in maize, conducted in both controlled environments and field conditions [15]. In some cases, GWAS may not yield the expected number of linked SNPs, which could be due to limited genetic variation in the population or limitations in sequencing technology resolution. In such instances, WGS may be the preferred approach. In conclusion, GWAS have been instrumental in identifying candidate genes and SNPs related to various stress resistances in maize, both biotic and abiotic. These findings hold promise for the development of stress-tolerant maize cultivars through molecular breeding and provide valuable insights into the underlying molecular mechanisms.

## 7. Identification of SNPs, QTL, and Candidate Genes for Trait Improvement

In recent decades, an extensive body of research has been dedicated to conducting GWAS to pinpoint QTL, SNPs, and candidate genes associated with complex targe traits, not only in maize but also in various other crops [122,123,124], which offer valuable insights for trait improvement. GWAS have identified thousands of genomic regions linked to various agronomic traits in plants, which are needed for the functional validation of allelic variants, and many GWAS on different plants, including Arabidopsis, rice, wheat, maize, soybean, etc., are reviewed [19,122,125,126,127]. Moreover, GWAS is being carried out not only on cereal crops but also on a wide range of crops, including cotton [128,129,130], tomato [131,132,133], sesame [45], peanut [134], lettuce [135], and peach [136]. Collectively, these studies combined with a purpose-developed population, database of allelic variation, and genotype–phenotype association offer resources for comprehending crops’ functional genomics, in addition to confirming previously validated trait association and also haplotypes. Here, we highlight key publications from 2018 to 2023 that have contributed to GWAS in maize (Table 1). For instance, the first candidate gene identified by GWAS, fatty acid desaturase 2 (fad2), revealed that changes in oleic acid content in maize grains result from allelic variations within the fad2 gene’s 5’ untranslated region [15,36]. Furthermore, GWAS has associated three candidate genes, ACP, COPII, and LACS, with oil content in maize kernels, along with four other genes [76].

Crucial maize plant architecture traits, such as plant height, ear height, leaf length, number of tassel branches, main axis length, and root architecture, significantly impact kernel yield. Recent studies have identified 189 candidate genes and 63 loci associated with root architecture in maize [137]. Additionally, two candidate genes, *Zm00001d018617* and *Zm00001d02365*, encoding gibberellin 2 oxidase and auxin factor 2, have been linked to plant height and the optimization of maize-stalk characteristics for improved biofuel production [138].

In the context of nutrient content, several SNPs, including S3_40522792, S2_1926586, S9_151265550, S3_186200393, S4_161165956, and S4_167189737, have been validated and associated with kernel zinc and iron content in maize [39,139,140]. GWAS have also revealed 32 SNPs significantly associated with maize lethal necrosis [141] and 44 SNP markers linked to hypersensitive defense reactions in maize [142]. Furthermore, the gene *Zmm22* has been implicated in vegetative growth, stalk diameter, and plant height [35], and it also plays a role in flowering time and reproductive transition in maize [143]. Another study integrated GWAS and co-expression analysis, identifying two genes, *Zm00001d002266* and *Zm00001d049584*, that regulate seedling root length in response to drought stress in maize [144].

Similarly, integrating GWAS and RNA-seq data, two candidate genes, *Zm00001d04319* and *Zm00001d039219*, have been associated with cold-stress responses during maize germination [145]. Additionally, 46 SNPs and 29 genes have been linked to grain-quality traits through GWAS [75]. Recent research has unveiled 27 candidate genes significantly involved in husk senescence, with key functions in husk senescence, husk morphogenesis, and responses to abiotic stress [146]. Collectively, a plethora of genes have been identified through QTL and GWAS analyses in various maize populations [147], facilitating gene cloning via diverse approaches.

**Table 1 ijms-25-01918-t001:** GWAS-based identified SNPs/QTL candidate genes that contribute to maize improvement.

Phenotypes Traits	Population	Sample Size	SNPs/QTL/Genes	Chromosomal Location	References
Ear traits (ear length, diameter, kernel length and width, cob diameter)	Inbred association population	292	20 SNPs	1, 2, 3, 4, 5, 6, 7, 8, 9, and 10	[124]
Corn earworm resistance	Diverse inbreed lines	287	51 SNPs	1, 2, 3, 4, 5, 6, 7, 8, 9, and 10	[103]
Root architecture traits	Diverse inbred lines	300	19 SNPs	1, 2, 5, 7, and 8	[103,148]
Gray leaf spot resistance	Diverse inbred lines	157	7 SNPs	1, 2, 3, 4, 5, 6, 7, and 10	[149]
Leaf angle and leaf orientation	diverse inbred lines	80	33 SNPs	1, 3, 4, 5, 6, 7, and 9	[150]
Male inflorescence morphology	Nested association mapping population	942	242 SNPs	1, 4, and 6	[151]
Starch pasting properties	Diverse inbred lines	230	60 QTNs	1, 2, 3, 4, 5, 6, 7, and 8	[152]
Tocopherol content	Diverse inbred lines	208	32 SNPs and 4 candidate genes	Multiple chromosomes	[153]
Stalk lodging resistance	Diverse inbred lines	257	423 QTNs and 63 candidate genes	1, 2, 3, 5, 6, 8, and 9	[154]
Southern corn rust resistance	Diverse inbred lines	253	7 SNPs	4, 8, and 10	[155]
Corn ear rot resistance	Diverse inbred lines	242	5 candidate genes	5, 7, and 10	[156]
Ear rot resistance	Diverse inbred lines	244	8 candidate genes	1, 2, 3, 5, 7, and 9	[107]
Fumonisin accumulation in kernels	Diverse inbred lines	270	39 SNPs/17 QTL	3 and 4	[157]
Stalk anatomy and stalk biomass	Diverse inbred lines	492	16 candidate genes	Multiple chromosomes	[35]
Plant architecture (plant height, leaf length and width and leaf angle	Diverse inbred lines	87	36 QTL		[158]
13 seedling traits under low phosphorus stress	Diverse inbred lines	356	551 SNPs	1, 2, 3, 4, 5, 6, 7, 8, 9, and 10	[159]
Plant height	Maize hybrids	300	9 SNPs and 2 candidate genes	1, 2, 4, 7, 9, and 10	[138]
Tassel architecture	Association panel	359	55 candidate genes/19 QTL	1, 2, 3, 4, 5, 6, 7, 8, 9, and 10	[160]
Popping expansion	Diverse inbred lines	183	4 SNPs		[161]
maize lethal necrosis (MLN) and Maize chlorotic mottle virus (MCMV)	Three double-haploid populations	965	54 SNPs and 40 QTL	1, 2, 3, 4, 5, 6, 7, 8, and 9	[162]
Goss’s wilt	NAM population	515	10 SNPs and 8 candidate genes		[163]
Salt tolerance	Diverse inbred lines	150	7 SNPs and 8 candidate genes	1, 3, and 6	[164]
Drought tolerance	Diverse inbred lines	210	26 QTL promising candidate genes	1, 2, 5, 8, and 10 3	[165]
Thermos tolerance of seed	Diverse inbred lines	261	4 QTL, 17 candidate genes and 42 SNPs	1, 2, 3, 4, 5, 6, 7, 8, 9, and 10	[116]
Grain yield and flowering time	Inbred association panel	300	1549 SNPs and 46 candidate genes	1, 2, 4, 5, 8, and 10	[166]
Husk tightness	Diverse inbred lines	508	27 candidate genes	1, 2, 3, 5, 6, 7, 8, and 10	[146]
Kernal row number	Diverse inbred lines	639	49 candidate genes and	1, 2, 3, 5, 9, and 10	[167]
Agronomic traits	Inbred lines	513	3 SNPs	4, and 3	[168]
Striga resistance	White maize inbred lines	132	24 SNPs	1, 3, 4, 5, 7, 8, 9, and 10	[169]
Maize leaf necrosis resistance	Diverse inbred lines	1400	32 SNPs and 9 candidate genes	1, 3, 4, 7, 9, and 10	[141]
Low nitrogen tolerance	Diverse hybrid lines	49	7 candidate genes	Multiple chromosomes	[170]
Agronomic traits	Inbred association lines	224	97 candidate genes and 73573 eQTL	1, 2, 3, 4, 5, 6, 7, 8, 9, and 10	[171]
Seminal root length	Inbred association lines	209	7 candidate genes	-	[144]
Low temperature	Diverse inbred lines	222	30 SNPs and 82 candidate genes	Multiple chromosomes	[145]
Leaf cuticular conductance	Diverse inbred lines	468	9 SNPs and 7 candidate genes	1, 4, 7, 8, and 10	[172]
Yield related traits	Double haploid population	250	138 SNPs, 100 QTL, and 52 candidate genes	1, 2, 3, 4, 5, 6, 7, 8, 9, and 10	[173]
Root architecture system	Diverse inbred lines	380	87 SNPs and 77 candidate genes	Multiple chromosomes	[174]
Fusarium verticillioides resistance	Maize association population	230	42 SNPs and 25 candidate genes	1, 2, 3, 4, 5, 6, 7, 8, 9, and 10	[175]
Corn leaf blight	Association mapping panel	419	22 SNPs	1, 6, 7, 8, 10	[176]
Aspergillus flavus resistance in kernels	Diverse inbred lines	313	4 SNPs and 16 candidate genes	1, 2, 8, and 9	[104]
Fusarium ear rot resistance	Diverse inbred lines	508	34 SNPs	-	[177]
Gray leaf spot resistance	Diverse inbred lines	410	22 SNPs	1, 2, 6, 7, and 8	[178]
Agronomic traits	Elite inbred lines	350	129 SNPs	1, 2, 3, 4, 5, 6, 7, 8, 9, and 10	[179]
Aboveground dry matter	Diverse inbred lines	412	129	1, 2, 3, 4, 5, 6, 7, 8, 9, and 10	[84]
Stover yield	MAGIC population	408	13 SNPs	-	[180]
Root architecture traits	RILs population	179	8 SNPs	1, 2, 4, and 10	[181]
Grain quality traits	Diverse inbred lines	248	49 SNPs and 29 candidate genes	1, 2, 3, 4, 5, 6, 7, 8, 9, and 10	[75]
Root hair length	Diverse inbred lines	281	11	1, 2, 4, 5, 6, and 10	[182]
Cold tolerance	Diverse inbred lines	80	4 SNPs and 12 QTL, 1 gene	3	[183]
Heavy metal stress	Double haploid lines	187	15 QTL and 4 genes	1, 2, 4, 7, and 10	[184]
Heat tolerance	Double haploid lines	662	46 SNPs	1, 2, 3, 6, 7, and 8	[185]
Cadmium toxicity	Diverse inbred lines	513	12 SNPs and 1 candidate genes	2	[186]
Seedling germination traits	MAGIC population	420	28 SNPs	2, 4, 5, 6, 7, 8, and 9	[187]
Grain yield and related traits	Inbred association panel	309	22 SNPs	-	[188]
Accumulation of micronutrients (Fe, Zn, Cu, Mn)	Diverse inbred lines	305	36 SNPs and 11 candidate genes	2, 3, 4, 6, and 8	[189]
Salt tolerance	Inbred association panel	305	120 candidate genes	-	[73]
Kernel moisture and dehydration rate	Diverse inbred lines	132	334 QTNs	2, 3, 4, 5, 8, and 9	[190]
Root traits	Diverse inbred lines	319	559 SNPs	Multiple chromosomes	[191]
Leaf angel	Diverse inbred lines	285	96 SNPs	1, 2, 3, 4, 5, 6, 7, 9, and 10	[192]
Root system architecture	Diverse inbred lines	421	63 SNPs and 189 candidate genes	1, 2, 3, 4, 5, 6, 7, 9, and 10	[137]
Grain yield quality traits	Association mapping population	410	42 SNPs	1, 2, 3, 4, 5, 6, 7, 8, 9, and 10	[193]
Yield related traits	Diverse inbred lines	291	59 SNPs and 66 candidate genes	1, 2, 3, 4, 6, 7, 8, 9, and 10	[79]
Grain yield and other traits	Diverse inbred lines	169	40 SNPs and 6 candidate genes	1, 2, 8, and 10	[194]
Metaxylem vessel brace roots	Association mapping panel	508	9 SNPs and 5 candidate genes	2, 4, 7, 8, and 10	[195]
Brace root	Association mapping panel	508	6SNPs and 27 candidate genes	3, 4, 5, 8, 9, and 10	[196]
Kernal related traits	Association panel	205	139 SNPs and 15 candidate genes	1, 2, 3, 5, 6, 7, and 9	[80]
Seed germination traits	Diverse inbred lines	321	58 SNPs	1, 4, 5, 6, 8, 9, and 10	[197]
Stalk lodging resistance	Diverse inbred lines	248	85 SNPs	1, 2, 3, 4, 5, 6, 7, 8, 9, and 10	[198]
Drought and heat resistance	Diverse inbred lines	162	117 SNPs and 20 candidate genes	1, 2, 5, and 7	[199]
Heat resistance	Diverse inbred lines	375	14 SNPs	1, 2, 4, 5, and 9	[200]
Rough dwarf disease resistance	Diverse inbred lines	292	22 SNPs	1, 3, 4, 7, and 8	[109]
Alkaline stress resistance	Association panel	200	9 SNPs	3, 4, 5, 6, and 9	[201]
Stalk sugar content and agronomic traits	Diverse inbred lines	188	92 SNPs	1, 3, 4, 6, 7, 8, and 10	[202]
Quality traits and starch pasting	Diverse inbred lines	292	48 SNPs 37 candidate genes	1, 3, 4, 5, 6, 7, 8, 9, and 10	[83]
Chlorophyll content	Diverse inbred lines	378	19 SNPs	2, 4, 5, 6, and 10	[203]
Chlorophyll content	Diverse inbred lines	290	140 QTNs and 11 key genes	-	[204]
Ear diameter	Multiple parent population	162	11 SNPs and 3 QTL	1, 2, 3, 6, 8, and 9	[205]
Stalk strength	Diverse inbred lines	345	94 QTL and 241 SNPs	1, 2, 3, 4, 5, 6, 7, 8, 9, and 10	[206]
Chilling tolerant	Diverse inbred lines	190	26 SNPs and 37 candidate genes	4, 6, 8, and 9	[119]
Striga resistance	Diverse inbred lines	141	22 SNPs	1, 3, 4, 5, 6, 7, 8, 9, and 10	[123]
Root hair length	Association panel	200	88 QTL	1, 2, 3, 4, 5, 6, 7, 8, 9, and 10	[207]
Total root length	Diverse inbred lines	280	38 candidate genes	1, 2, 3, 4, 6, 7, 8, and 9	[208]
Root morphology and phosphorus acquisition	Diverse inbred lines	561	7 SNPs	8	[209]
Leaf streak resistance	Diverse inbred lines	200	11 SNPs	1, 2, 5, 7, 8 and 9	[210]
Drought resistance	Association panel	379	15 candidate genes	1, 3, 4, 5, 6, 8, and 9	[211]

## 8. Pervasive Pleiotropy in Maize GWAS Studies

In the realm of maize research, numerous GWAS have unearthed thousands of SNPs and candidate genes linked to various phenotypic traits, a phenomenon commonly referred to as pleiotropic SNPs and candidate genes. Pleiotropy, in genetics, occurs when one or more seemingly unrelated phenotypic traits are controlled by a single gene, a concept that has garnered considerable attention in maize research [212]. Pleiotropy serves as a genetic nexus connecting economically and agriculturally relevant traits, and it can also be detected through linkage disequilibrium, adding a layer of complexity to our understanding of maize genetics [213]. Numerous research studies have led to the discovery of pleiotropic genes in maize, governing multiple aspects of maize crops. In this discussion, we delve into the potential implications and underlying causes of pleiotropic SNPs and genes.

One proposed explanation for pleiotropy is that a single gene may be involved in various cell types or take part in signaling cascades targeting multiple endpoints. It can be challenging to differentiate between genuine biological pleiotropy and mediated or spurious pleiotropy. This is because genes typically function in complex pathways and networks, leading to interconnected traits. Mediated pleiotropy arises when two linked phenotypes share a common pathway, whereas spurious pleiotropy occurs when a detected SNP is located within a small region of high linkage disequilibrium (LD), encompassing two closely adjacent but distinct genes, each regulating different traits. Despite the wealth of GWAS studies, an in-depth understanding of pleiotropic genes remains elusive. However, we can consider pleiotropic genes as promising targets for precise genome editing techniques, such as the CRISPR/Cas9 system. Such tools allow us to explore the potential consequences of correlated phenotypes by fine-tuning specific genomic regions containing pleiotropic genes in maize. Editing these genes can collectively influence metabolic processes, potentially avoiding the need to target multiple genomic regions individually. Furthermore, modifying pleiotropic genes could lead to the emergence of novel phenotypes with significant effects.

While we’ve discussed several GWAS investigations that have identified pleiotropic genes influencing seemingly unrelated traits, it is important to note that comprehensive validation of pleiotropic genes discovered via GWAS is lacking. As more GWAS reports continue to emerge, maize breeders are likely to pay increased attention to pleiotropic genes. However, it is worth noting that there is currently limited research on the functional validation of pleiotropic genes or SNPs identified through GWAS [20]. In the future, with the development of new statistical models and genomic tools for GWAS studies, we may gain further insights into pleiotropic loci. As researchers increasingly focus on pleiotropy, the growing number of phenotypic–genotypic associations will likely expand our understanding of this intriguing genetic phenomenon.

## 9. Benefits and Limitations of GWAS

Over the past decade, GWAS have sparked a revolution in our understanding of the intricate genetic architectures underlying major agronomic traits. These studies have forged robust connections between genetics and complex quantitative traits, contributing significantly to our comprehension of maize functional genomics and genetics. Notably, GWAS has led to the discovery of novel genes and biological pathways, propelling successful molecular breeding programs across various crops [3,15]. While GWAS have been a powerful tool, it is not without its controversies. The following discussion will delve into both the benefits and limitations of GWAS.

### 9.1. Benefits of GWAS

Integration of genotype and phenotype: GWAS is a potent method for seamlessly integrating genotype and phenotype data, enhancing our understanding of complex traits.Identification of causal and predictive factors: It has the capability to pinpoint both causal and predictive factors associated with specific traits, allowing for in-depth genetic analysis.Applicability: GWAS can be conducted on breeding populations as well as natural populations, broadening its utility.Discovery of novel associations: It has successfully uncovered novel associations between genetic variants and traits, expanding the scope of genetic research.Pathway independence: Unlike QTL mapping, GWAS does not require prior knowledge of the biological pathways related to the studied traits, enabling the discovery of new biological mechanisms.Candidate gene discovery: GWAS can identify previously unidentified candidate genes, contributing to the expansion of genetic knowledge.Collaboration promotion: GWAS encourages collaborative consortia, facilitating the recruitment of a sufficient number of participants for robust analyses and fostering continued collaboration.Ancestry data: It provides ancestry data for each subject, aiding in matching case and control subjects, ensuring the reliability of the analysis.Structural variant consideration: GWAS takes into account two types of structural variants sequence variation and copy number, yielding more comprehensive and reliable data.Complex trait understanding: It is well suited for unraveling the genetic contributors to complex traits, where an individual’s genes may have a minor influence.Data availability: GWAS data is often made publicly available, facilitating the discovery of new trait association and promoting transparency.Ethnic diversity: GWAS can shed light on ethnic differences in complex traits, contributing to a more comprehensive understanding of genetic diversity.

### 9.2. Limitations of GWAS

Significance threshold: A major limitation is the need for a stringent significance threshold to account for multiple test burdens, potentially missing important associations. Statisticians are strict about this, but if you can prove it with biological evidence, it is not a problem.Low-frequency variant analysis: GWAS is not appropriate for studying low-frequency and rare variants. When this happens, a parental population needs to be constructed to detect this rare variant.Replication and population size: Findings must be replicated in independent samples from diverse populations, necessitating large and diverse study populations.Association vs. causation: GWAS identifies associations but does not pinpoint causal variants and genes. Candidate gene selection and its biological validation are necessary.Specific site identification: It may identify specific genetic sites rather than entire genes, and many identified variants are not directly linked to protein-coding regions.Missing heritability: GWAS cannot elucidate all genetic determinants of complex traits, leaving much of the heritability unaccounted for.Molecular biology insights: Findings related to GWAS variations do not necessarily reveal the underlying molecular biology of traits. Biological validation is necessary.Ongoing challenges: While technology, computing, methodology, population stratification, and whole genome sequencing (WGS) may address some limitations, challenges persist in achieving a comprehensive understanding of complex traits.

In conclusion, despite a substantial number of GWAS studies conducted in the last decade, the ongoing increase in such studies attests to their success in uncovering the genetic basis of complex plant characteristics. Recognizing and addressing the limitations of GWAS remain pivotal for future advancements in this field.

## 10. GWAS Interpretation with OMICS

GWAS have emerged as crucial tools for unraveling the genetic architecture of complex traits. Over the last few years, GWAS have led to the discovery of thousands of genetic loci and candidate genes associated with diverse complex traits, including responses to abiotic and biotic stress [3,15,59,75,214]. The analysis of maize populations through GWAS has been on the rise, providing valuable insights into the genetic basis of complex traits. However, interpreting GWAS findings remains challenging, as they do not inherently provide information about the underlying biological or environmental factors influencing these traits [214,215]. To enhance maize development, especially in terms of resistance to environmental stress, it is essential to gain a deeper understanding of the genetic architecture governing complex traits [118,139,197,216]. While GWAS identifies genetic associations, it is limited in elucidating the intricate mechanisms and environmental influences driving these associations. In recent years, a promising approach to complement GWAS and gain a more comprehensive perspective on complex traits is the integration of multi-omics data [217]. This approach combines genomics, epigenomics, transcriptomics, proteomics, metabolomics, and advanced statistical methods to provide a holistic view of the molecular mechanisms underlying complex traits, including responses to abiotic and biotic stress, as well as yield improvement [218]. Integrating multi-omics data offers several advantages, for example, it allows researchers to delve deeper into the molecular mechanisms governing complex traits, shedding light on the underlying biological processes. Multi-omics integration considers not only genetic factors but also environmental influences, providing a more complete picture of trait variation. It aids in the identification of potential genes and their associated pathways, facilitating a more nuanced understanding of trait regulation.

In conclusion, the integration of multi-omics data with GWAS holds tremendous potential for advancing our understanding of complex traits in maize. This approach enables researchers to explore the intricate molecular networks that govern responses to environmental stress and yield improvement. By combining genetic, epigenetic, transcriptomic, proteomic, and metabolomic information, we can gain a more holistic view of the factors shaping complex traits, ultimately contributing to the development of more resilient and productive maize varieties.

## 11. Conclusions and Future Prospects in Maize

Maize, as a pivotal cereal crop, exhibits remarkable adaptability to diverse environmental conditions. Recent advancements in maize whole genome sequencing and resequencing have paved the way for the identification of millions of genome-wide SNPs, QTL, and candidate genes. This wealth of genetic information stems from a wide array of naturally occurring variants gathered across different environments, reflecting years of genetic diversity accumulation. These comprehensive studies have been instrumental in mapping genes governing critical aspects of maize, such as yield and its associated traits, tolerance to biotic and abiotic stresses, and quality attributes. However, it is paramount to validate the candidate genes and loci linked to specific traits, and the development of genomic resources is pivotal in shaping the landscape of whole-genome prediction models.

GWAS have emerged as a powerful tool for predicting allele functions, pinpointing mutations, and identifying candidate genes responsible for desired agronomic traits. The advent of DNA sequencing has opened the door to deep analyses of natural variations within plant genomes, allowing researchers to harness these resources in conjunction with GWAS to elucidate the genetic underpinnings of complex features. Recent breakthroughs in quantitative omics technologies such as transcriptomics, metabolomics, and epigenomics have introduced innovative association studies like Metabolite-Wide Association Studies (MWAS), Epigenome-Wide Association Studies (EWAS), and Transcriptome-Wide Association Studies (TWAS). These holistic approaches augment the capabilities of GWAS, offering valuable insights into the genes underlying agriculturally significant characteristics and expediting genomics-assisted breeding.

To thrive in this evolving landscape, it is imperative to develop more efficient GWAS computational techniques. A deeper comprehension of genetic variability at the SNP level will prove invaluable in conserving, characterizing, and exploiting diverse germplasm. GWAS, in turn, will enrich breeding programs by expanding access to desirable agronomic traits and germplasm collections.

However, it is worth noting that a substantial portion of research has primarily focused on the primary (additive) effects of genetic architecture. To comprehensively understand the complexity of complex traits, which encompass gene networks, epigenetic influences, interactions with the environment, and rapidly changing conditions, a multifaceted approach is needed. Potential solutions include acquiring pertinent phenotypic data, refining statistical models, and verifying potential loci, all of which can expedite molecular breeding strategies in maize. While GWAS remains indispensable with the integration of next-generation sequencing (NGS) technology, advancements in statistical methodologies and genomic designs hold the key to enhanced effectiveness.

Careful consideration should be given to the selection of analytical methodologies, with an option to combine complementary methods when tackling intricate genetic traits. Moreover, improved population designs, core collection techniques, novel sequencing approaches, and statistical methods may unveil and manipulate genetic factors contributing to quantitative variation. In this evolving landscape, innovative designs can enhance precision and accuracy by reshaping allelic spectra and minimizing confounding variables. Moreover, it is imperative that candidate genes identified through GWAS undergo biological validation. For a more interesting traits, peaks identified by a new GWAS can be compared with known genomic region or related genes for validation [219]. Utilizing transgenic and alternative methods, such as RNA interference, mutant validation, gene knockout, overexpression, and CRISPR/Cas9-mediated gene silencing, enables the exploration of new causal genes underlying GWAS peaks [19,220,221]. In conclusion, we advocate for the integrated use of muti-omics data alongside genetic design and appropriate analytical techniques, as they hold the promise of uncovering the biological foundations of phenotypic variation in maize molecular breeding programs.

## 12. Provoking Questions in GWAS

How can we achieve a high-resolution identification of causal genes and SNPs in order to fully appreciate the role of pleiotropy in GWAS?

How might the insights gained from GWAS be applied in the near future to enhance the genetics of crop plants?

How close are we to effectively developing climate-smart cultivars by harnessing natural variations uncovered through GWAS?

When a trait of interest is influenced by a rare variant, why is that trait prevalent in a large population, while GWAS struggle to uncover the associated rare variant?

Can the study of genes and processes that underlie phenotypic and physiological changes enable us to predict how crops will respond to ever-changing environmental conditions?

## Figures and Tables

**Figure 1 ijms-25-01918-f001:**
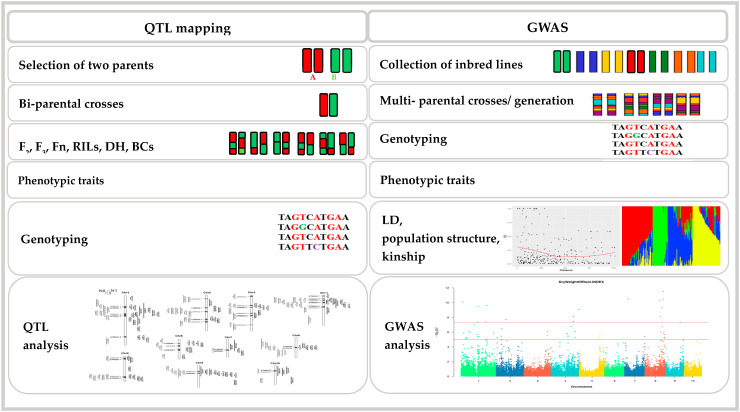
General difference between QTL mapping and GWAS.

**Figure 2 ijms-25-01918-f002:**
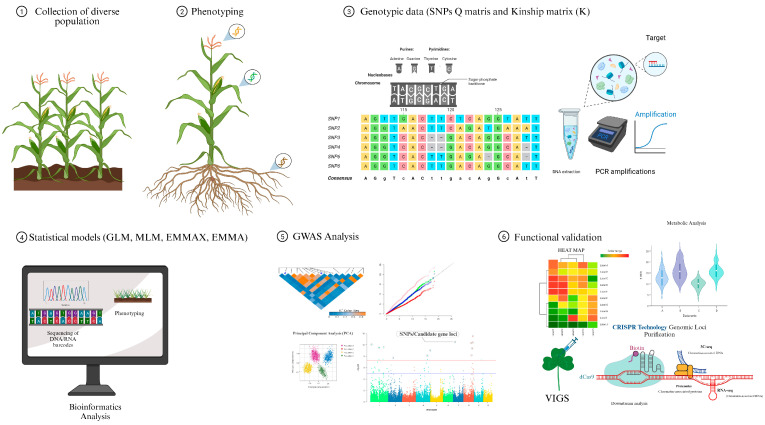
Basic GWAS approach in maize. (1) Collection of diverse maize population. (2) Phenotyping. (3) Genotyping. (4) Statistical models. (5) Identification of significant SNPs. (6) Functional analysis of candidate genes associated with phenotyping.

**Figure 3 ijms-25-01918-f003:**
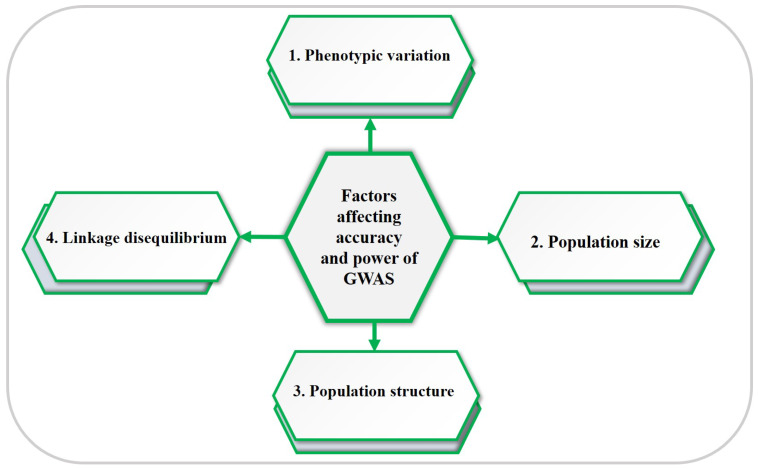
General factors affecting the accuracy and resolution power of GWAS.

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
