# Peer review of "Advancements and Prospects of Genome-Wide Association Studies (GWAS) in Maize"

_ijms, 2024, doi:10.3390/ijms25031918_

Round 1

Reviewer 1 Report

Comments and Suggestions for Authors

enclose PDF

Author Response

We are very grateful and thankful to you for giving us the opportunity to revise the article. Please consider the revised version of our article “Advancements and Prospects of Genome-wide Association Studies (GWAS) in Maize” to be published in the journal of “IJMS”. In the following pages are our point to point responses to each of the comments raised by the Reviewer 1#. We hope that the revisions in the manuscript and our accompanying responses will be sufficient to make the manuscript suitable for publication in IJMS.

Sincerely,

Xuehai Zhang

Key Laboratory of Wheat and Maize Crops Science/College of Agronomy,

Henan Agricultural University, Zhengzhou 450002, P.R. China.

Email:[email protected]

Reviewer 1#:

Q1. “Advancements and Prospects of Genome-wide Association …..” by Sahito et al. (2023), was assessed by me. Although with a scientific level of 7-8 (out of a score of 10), the state of the art in Sec. 1 is acceptable, both in the text and in the references used. With respect to the second section: line 95: Genome-Wide Association Studies (GWAS) in Maize.  Line 114: …. 2017). That is, some genes do not segregate independently because the two loci involved are located on the same chromosome (include). LD, a measure….

R1: Thank you very much for your kind and careful reading. Please note the following changes in the revised version and revised text is highlighted in red color. Line number 105 to 122

Q2. Regarding sec. 3: reading several times this part of the Review gave me the feeling that I was examining the methodology of a paper. Which is not the purpose of an update (i.e. 134 lines).

R2: Thanks for your comments. We have deleted the methodology section in the revised manuscript. Thanks again for your suggestion.

Q3.  Lines 190-93: this sentence is damning. This is confirmed by the reference to the matter (Burgarth et al., 2017)

R3: Thanks for your comments. We have modified this sentence in revised manuscript and also confirmed by  (Burgarth et al., 2017). Line number 177 to 181

Q4. The title, "The Basic GWAS Approach", gave me bad omens. And as I indicated before, I confirmed it.

R4: Thank you very much for carefully reading, We have modified this section. Please kindly check in the revised manuscript. See Line number 148 to 217

Q5. Sec. 4: line 261: “low heritability can diminish the power of GWAS and the ability to detect associations”. line 271: “a smaller population size can diminish the power of GWAS”; Line 278: “population structure is a critical factor affecting GWAS …”; “Neglecting allele frequency can lead to misleading results” line 304: “LD decay rate is essential …”. These 5 aspects strongly affect the use of this technology in plants (in sec. “9.2.Limitations of GWAS”, the authors consider these weaknesses. That is, if a method is restricted to specific characteristics of the species to be studied..... Regulate promotion for GWAS. On the other hand, the authors use 235 lines to describe this technique in maize. Indeed, Zea mays belongs to the title of this Review. But, I ask: how many species of plants, apart from corn, have been studied with this technique? I would really appreciate it if you could dedicate some space to them in this update.

If so, this is not the case.

If the authors make an acceptable “point by point” I might reconsider my refusal to publish this paper.

R5: Thank you for your valuable comments, we have modified these five aspects and the “Limitations of GWAS”. section. In addition, GWAS studies have also been done in many plants, we have added relevant description in “1. Introduction” and the revised manuscript. Line numbers 69 to 96, 248 to 250, 259 to 261 , 268 to 295 and 580 to 601. 

Thank you again for your hard work and I have learned a lot from you, hope you can be satisfied with our revisions, and consider our article published.

Reviewer 2 Report

Comments and Suggestions for Authors

The manuscript “Advancements and Prospects of Genome-wide Association Studies (GWAS) in Maize” is a very interesting study. Just a few recommendations to the authors:

 1-      In the Introduction section, the Authors emphasize the issue of drought resistance; However, this theme is developed the same as other themes in the text. Therefore, Authors are recommended to balance the section with the themes developed in the manuscript.

2-      The Authors mention that in the manuscript they address GWAS prospects in maize; however, this issue is not often mentioned in the manuscript. Therefore, the Authors are recommended to address this issue more specifically.

3-      In lines 452-454 the Authors mention very briefly the validation of GWAS results in field, so the authors are recommended to comment more on this topic, as well as the applications in the field that these studies have had.

4-      Authors are recommended to review in the guide for authors how to cite references in the text and how to make the references section.

Author Response

We are very grateful and thankful to you giving us the opportunity to revise the article. Please consider the revised version of our article “Advancements and Prospects of Genome-wide Association Studies (GWAS) in Maize” to be published in the journal of “IJMS”. In the following pages are our point to point responses to each of the comments raised by the Reviewer 2#. We hope that the revisions in the manuscript and our accompanying responses will be sufficient to make the manuscript suitable for publication in IJMS.

Sincerely,

Xuehai Zhang

Key Laboratory of Wheat and Maize Crops Science/College of Agronomy,

Henan Agricultural University, Zhengzhou 450002, P.R. China.

Email:[email protected]

Reviewer 2#:

Q1. In the Introduction section, the Authors emphasize the issue of drought resistance; However, this theme is developed the same as other themes in the text. Therefore, Authors are recommended to balance the section with the themes developed in the manuscript.

R1: Thank you very much for your comments and suggestions. In order to balance the section with the themes, we have carefully checked and modified the introduction section according to your suggestion. We have revised them in the revised manuscript and highlighted them in red color. Please see line 40 to 47.

Q2. The Authors mention that in the manuscript they address GWAS prospects in maize; however, this issue is not often mentioned in the manuscript. Therefore, the Authors are recommended to address this issue more specifically.

R2:Thank you for your valuable comment and suggestion. We have mentioned this issue in the revised manuscript according your recommendation. Please see line 638 and 681 to 687.

Q3. In lines 452-454 the Authors mention very briefly the validation of GWAS results in field, so the authors are recommended to comment more on this topic, as well as the applications in the field that these studies have had.

R3: Thank you for your valuable comments and suggestions, we have added more on this topic as per your recommendation in the revised manuscript. Please see line 434 to 448, line 688 to 494.

Q4. Authors are recommended to review in the guide for authors how to cite references in the text and how to make the references section.

R4: Thank you for your careful reading, we have corrected the citation format of references in the reference section and the text according to the guide for authors of the journal of “IJMS”.

Round 2

Reviewer 1 Report

Comments and Suggestions for Authors

I agree with the majority of the paragraphs included in the secons version of this draft. However, it is neccesary to correct certain gramatical English errors. 

Comments on the Quality of English Language

gramatical errors